# OpenReview forum: "DeepSeek-R1 Thoughtology: Let’s think about LLM reasoning"
_TMLR — Accepted by TMLR_

### Review · Reviewer_GSwv · 2025-08-17

**Summary Of Contributions:**

This work thoroughly and empirically studies the reasoning process of deepseek, having a lot of interesting first-step observations that future works of the field could build on

It finds for example:
1. deepseek's behavior is different for different languages
2. the thinking budget cannot be controlled by prompting
3. deepseek is prone to overthinking

**Additional Comments:**

A question: the proposed taxonomy is never used again after section 3, right?

**Audience:**

Yes

**Audience Explanation:**

I think this is obvious. Any empirical study of LLMs are certainly of interest

**Broader Impact Concerns:**

No ethical concerns

**Claims And Evidence:**

Yes

**Claims Explanation:**

I think all the claims are backed up by numerical evidences, after all, none of the claims are very strong

**Requested Changes:**

Nothing much I feel. This is an interesting, comprehensive paper that decribes the reasoning behavior of deepseek. Of course, it does not go deep into any of the topics, which is a weakness that the authors have acknowledged. Overall, this is an interesting and useful read for me.

Figure 11.3 could be fixed -- green and red colors are not visible to color blind people

---

> ### Author Response · Authors · 2025-10-02
> **Response**
>
> We thank you for reviewing our paper. We appreciate your positive assessment.
>
> ### **Revisions**
>
> > *Figure 11.3 could be fixed -- green and red colors are not visible to color blind people*
>
> Thank you for the suggestion. We will change the colors in the plot in the final version to ensure accessibility of our findings.
>
> ### **Questions**
>
> > *the proposed taxonomy is never used again after section 3, right?*
>
> The taxonomy is not directly used but it underpins several of our investigations and is referenced in later sections with analyses over reconstruction cycles and rumination in Sections 5,6,9, and 10.
>
> We would also like to point the reviewer to the additional experiments we did focusing on quantifying rumination (see general response) which is a phenomenon observed in the reconstruction cycle(s) as defined in our taxonomy.

---

### Review · Reviewer_Q4pz · 2025-09-21

**Summary Of Contributions:**

The paper starts a new field called "Thoughtology" to study how reasoning models like DeepSeek-R1 "think" by looking at their reasoning steps. It finds that there is a "sweet spot" for how long the model should think. Thinking for too long can make its answers worse. The model often gets stuck thinking about the same ideas over and over again, which the paper calls "rumination." It shows that DeepSeek-R1 has serious safety problems and is less safe than its non-reasoning version. The model's cultural values and how long it thinks change significantly depending on whether it is prompted in English or Chinese. Its reasoning time increases for sentences that are also difficult for humans, but it also thinks in a non-humanlike way on simple sentences. In visual and physical reasoning tasks, the model fails to improve its work step-by-step and instead relies too heavily on abstract mathematical thinking. The model cannot control its thinking time based on user instructions, and training it to follow a budget may reduce its task accuracy.

Strengths:

- Comprehensive evaluation: The paper offers a broad and thoughtful assessment of DeepSeek-R1 across multiple dimensions—performance, safety, cognitive plausibility, and controllability—providing a holistic view of the model’s behavior.

- Rich qualitative insights: By leveraging the transparency of the model’s reasoning traces, the authors present compelling, concrete examples that illustrate complex phenomena such as knowledge conflicts, rumination, and non-humanlike reasoning. These qualitative cases greatly enhance the interpretability and credibility of the findings.

- Solution-oriented mindset: Beyond diagnosing problems, the paper takes a constructive approach. For instance, after demonstrating the model’s inability to adhere to reasoning time constraints, the authors propose and test a proof-of-concept reinforcement learning method to improve compliance—a step toward actionable solutions.


Weaknesses:

- Lack of cross-model comparison: The study focuses almost entirely on DeepSeek-R1, limiting the generalizability of its conclusions. Without comparing with other reasoning models—such as OpenAI’s O1/O3 series or Qwen3—it remains unclear whether the observed behaviors are specific to this architecture or common across large reasoning models. Including such comparisons would significantly strengthen the paper’s impact and relevance.

- Insufficient quantitative support for key concepts: While the idea of rumination is compelling and well-illustrated through examples, it remains largely descriptive. Introducing quantitative metrics—such as repetition rates in reasoning tokens or self-similarity scores over thought sequences—could make these analyses more rigorous and scalable.

- Preliminary nature of some analyses: As the authors acknowledge, this is an exploratory study. While the breadth is impressive, some investigations remain shallow due to limited sample sizes or narrow dataset coverage. Several findings point to promising directions for future work, but deeper dives into individual phenomena would further solidify the contributions.

**Audience:**

Yes

**Audience Explanation:**

The reasoning mechanisms of large language models are a compelling and worthwhile area of research.

**Broader Impact Concerns:**

None.

**Claims And Evidence:**

No

**Claims Explanation:**

Please see "Weaknesses".

**Requested Changes:**

Please see "Weaknesses".

---

> ### Author Response · Authors · 2025-10-02
> **Response**
>
> We thank you for reviewing our paper. We are pleased to see that you found our evaluation comprehensive and insightful. We address your comments below:
>
> ### **Response to Weaknesses**
>
> > W1. *Lack of cross-model comparison*
>
> Our paper focuses on DeepSeek-R1 because it was a major milestone for open-source research: it was the only widely-available ‘open-trace’ reasoning model during the time this work was carried out and submitted. Models like o1/o3 do not expose full chains of thought, and the “pure-thinking” Qwen3 variant was released after our submission. We’re open to adding some experimental results for Qwen3-thinking – if you consider it important for your assessment, we can include it in the next revision; otherwise, we’ll add it by the camera-ready.
>
> > W2. *Quantifying rumination*
>
> Thank you for this excellent suggestion. We have now incorporated quantitative metrics to measure rumination – please refer to the general response for a summary.
>
> > W3. *Preliminary nature of some analyses*
>
> Our aim in this study was to establish both a set of core, reproducible observations and a broad survey of behaviors across settings. Given space constraints, we do not pursue exhaustive analyses of every phenomenon. Several threads do receive deeper treatment, for example: reasoning-pattern analysis (section 3), thought-length analysis (section 4), psycholinguistic analyses (section 9), and efficiency/budget trade-offs (section 11). For some other threads, we document the recurring patterns with preliminary analyses with the hope to show the community that they are worth deeper inquiry. Our work provides a shared vocabulary that future work can build on, diving deeper into each phenomenon with specialized analysis methods and datasets. Future work can explore some of these phenomena and issues in depth, but the overarching findings we report in our paper are only possible with such a broad scope.

---

### Review · Reviewer_yXpG · 2025-09-26

**Summary Of Contributions:**

The paper carried out an extensive investigation into the reasoning capabilities of Language Reasoning Models (LRMs), using DeepSeek R1 as a case study. The investigation covered several areas, including analysis of the model itself (scaling thought length, thought structure/pattern, exploration of thoughts), LRMs as a cognitive engine (world modelling, language), and safety and cultural/ethical concerns. The paper maintains a balanced perspective while highlighting its limitations, which include limited depth in the areas investigated and limited evaluation of reasoning techniques (focusing only on DeepSeek R1 techniques).

**Strengths**
1. The paper is well structured.
2. Extensive breadth of areas investigated.

**Weaknesses**
1. Limited depth in analysis.
2. Use of single LRM approach.
3. The paper could benefit from a deeper investigation into why LRMs fail at visual reasoning and world modeling tasks.

**Question**
1. Page 3, bullet point 2: “Continuously scaling length of thoughts does not necessarily increase performance. There exists a problem-specific optimal reasoning length, beyond which performance declines.” Could you comment on why the performance only declines rather than plateau (stay the same)?

**Audience:**

Yes

**Audience Explanation:**

The study of Language Reasoning Models (LRMs) is an emerging sub-field of foundation models. This study provides a comprehensive analysis of one such LRM, with results that could serve as a foundation for future research by the TMLR audience.

**Claims And Evidence:**

Yes

**Claims Explanation:**

The claims are supported by empirical results, both quantitative and qualitative.

**Requested Changes:**

For the analysis conducted in section 9 (LRMs as cognitive engines), is it possible that the dissimilarity between human reasoning and LRM reasoning in the garden path and comparative illusion tasks stems from the structured pattern of reasoning (problem decomposition) used by DeepSeek-R1 (which is better suited to traditional reasoning multi-step tasks like math)? In essence, such an approach to reasoning could have been encoded as a prior in the base model or acquired as a result of the type of data on which it was trained.

---

> ### Author Response · Authors · 2025-10-02
> **Response**
>
> We thank you for reviewing our paper. We are pleased with your positive assessment and glad that you found our paper extensive and well-structured. We address your comments below:
>
> ### **Response to Weaknesses**
>
> > W1. *Limited depth in analysis.*
>
> Our aim in this study was to establish both a set of core, reproducible observations and a broad survey of behaviors across settings. Given space constraints, we do not pursue exhaustive analyses of every phenomenon. Several threads do receive deeper treatment, for example: reasoning-pattern analysis (section 3), thought-length analysis (section 4), psycholinguistic analyses (section 9), and efficiency/budget trade-offs (section 11). For some other threads, we document the recurring patterns with preliminary analyses with the hope to show the community that they are worth deeper inquiry. Our work provides a shared vocabulary that future work can build on, diving deeper into each phenomenon with specialized analysis methods and datasets. Future work can explore some of these phenomena and issues in depth, but the overarching findings we report in our paper are only possible with such a broad scope.
>
> > W2. *Use of single LRM approach.*
>
> Our paper focuses on DeepSeek-R1 because it was a major milestone for open-source research: it was the only widely-available ‘open-trace’ reasoning model during the time this work was carried out and submitted. Models like o1/o3 do not expose full chains of thought, and the “pure-thinking” Qwen3 variant was released after our submission.
>
> > W3. *The paper could benefit from a deeper investigation into why LRMs fail at visual reasoning and world modeling tasks.*
>
> We agree this is an important (and difficult) research question, but a full causal account is beyond the scope of this study. We believe there could be two likely contributors: (i) data mismatch: post-training corpora focusing on reasoning contain relatively little grounded visuospatial/physics supervision compared to math, QA, etc., and (ii) update mechanics: the model does not iteratively edit a latent scene/world state, so “thinking longer” doesn’t yield step-wise improvements. Our goal here was to document the phenomenon. Pinpointing causes with more confidence will require targeted evaluations and ablations, which we flag as high-value future work.
>
> ### **Questions**
>
> > *why the performance only declines rather than plateau (stay the same)?*
>
> In the paper, we provide two plausible explanations for why overly long thoughts hurt performance (page 17): (1) the model goes down the wrong path towards solving the problem and then it keeps trying unsuccessfully until it decides to give up, never quite finding the correct approach (as shown in Figure B.2); and (2) the model finds the correct approach and solution but then self-verifies it to be incorrect, eventually outputting a different incorrect answer (as shown in Figure B.3). Moreover, our new experiments quantifying rumination (Fig. 4.4) indicate that it may also be a factor negatively affecting performance.
>
> ### **Requested Changes**
>
> > *is it possible that the dissimilarity between human reasoning and LRM reasoning in the garden path and comparative illusion tasks stems from the structured pattern of reasoning (problem decomposition) used by DeepSeek-R1 (which is better suited to traditional reasoning multi-step tasks like math)?*
>
> Indeed yes, we agree with you and have added a note regarding this in our revision (end of section 9.3).

---

### Author Response · Authors · 2025-10-02
**General Response and Summary of Revisions**

We thank all the reviewers for spending their valuable time reviewing our paper. We’re encouraged by the recognition of the paper’s clear structure (yXpG), its extensive breadth and comprehensive scope (yXpG, Q4pz, GSwv), the interesting insights it surfaces (GSwv, Q4pz), and the constructive, actionable framing of our findings (Q4pz). Below, we provide a summary of revisions:

### **Measuring rumination**

As suggested by reviewer Q4pz, we have now carried out additional experiments to quantify rumination:

In section 3.3, we define rumination rate using two measures, (i) n-gram repetition rate, and (ii)  lexical diversity. We show that rumination rises with task complexity (Fig. 3.6), and that it varies widely across different tasks and is independent from simply processing time or time spent in reconstruction cycles (Fig 3.3b).

In section 4, we show that incorrect responses have a higher rumination rate compared to correct responses for math reasoning tasks (Fig. 4.4), indicating that rumination may hurt performance and might be one of the reasons why overly long CoTs fail to reach the correct answer.

In section 9, we show that rumination is higher for psycholinguistic stimuli vs controls (Fig. 9.3).

### **Note discussing LRM and human reasoning dissimilarity**

As suggested by reviewer yXpG, we have added a note at the end of section 9.3 speculating about a potential reason behind the divergence in human reasoning and LRM reasoning for the garden path and comparative illusion experiments.


### **More samples in long-context reasoning analysis**

As suggested by reviewer Q4pz, we experiment with increased sample size for long-context reasoning analysis on the CHASE dataset in section 5.2 which strengthens our findings.

---

### Author Response · Authors · 2025-10-06
**Request**

Dear reviewers, we have provided clarifications and responses to your comments, and revised the paper with additional experimental results. Could you kindly look at our rebuttal and revision and let us know if your concerns have been addressed? Thanks!

---

### Decision · Action_Editor_MA27 · 2025-12-09

**Recommendation:** Accept as is

**Audience:**

Yes

**Audience Explanation:**

The paper is a bout LLM reasoning which is a relevant topic for TMLR.

**Claims And Evidence:**

Yes

**Claims Explanation:**

The paper provides an empirical study for the reasoning process of DeepSeek-R1. The results are supported by clear, accurate, and convincing evidence, as required by the TMLR criteria. Some notation findings (as noted by one of the reviewer) show that deepseek's behavior is different for different languages, its "thinking" time/budget cannot be controlled by prompting, and that it can "overthink" a problem where results can get worse given more time.

All reviewers agree that the paper is relevant and should be accepted. The paper however does not go deep into any of the topics which is a weakness acknowledged by the authors. It does not make the paper any less relevant but it is worth noting.